# Self-Sanitizing Polycaprolactone Electrospun Nanofiber Membrane with Ag Nanoparticles

**DOI:** 10.3390/jfb14070336

**Published:** 2023-06-25

**Authors:** Elizaveta S. Permyakova, Anton Manakhov, Philipp V. Kiryukhantsev-Korneev, Anton S. Konopatsky, Yulia A. Makarets, Kristina Yu. Kotyakova, Svetlana Yu. Filippovich, Sergey G. Ignatov, Anastasiya O. Solovieva, Dmitry V. Shtansky

**Affiliations:** 1National University of Science and Technology “MISIS”, Moscow 119049, Russia; kiruhancev-korneev@yandex.ru (P.V.K.-K.); ankonopatsky@gmail.com (A.S.K.); jl.makarets@gmail.com (Y.A.M.); kristinkagudz@mail.ru (K.Y.K.); shtansky@shs.misis.ru (D.V.S.); 2Research Institute of Clinical and Experimental Lymphology—Branch of the ICG SB RAS, 2 Timakova st., Novosibirsk 630060, Russia; solovey_ao@mail.ru; 3Bach Institute of Biochemistry, Moscow 115419, Russia; syf@inbi.ras.ru; 4State Research Center for Applied Microbiology and Biotechnology, Obolensk 142279, Russia; ignatov.sergei@gmail.com

**Keywords:** polycaprolactone nanofibers, plasma polymerization, Ag nanoparticles, antibacterial activity, antifungal activity, face masks

## Abstract

The objective of this research was to develop an environment-friendly and scalable method for the production of self-sanitizing electrospun nanofibers. This was achieved by immobilizing silver nanoparticles (Ag NPs) onto plasma-treated surfaces of biodegradable polycaprolactone (PCL) nanofibers. The plasma deposited polymer layer containing carboxyl groups played a critical role in providing a uniform distribution of Ag NPs on the nanofiber surface. Ag ions were absorbed by electrostatic interaction and then reduced under the action of UV-light. The concentration and release of Ag ions were analyzed using the EDXS/XPS and ICP AES methods, respectively. Although high levels of Ag ions were detected after 3 h of immersion in water, the material retained a sufficient amount of silver nanoparticles on the surface (~2.3 vs. 3.5 at.% as determined by XPS), and the release rate subsequently decreased over the next 69 h. The antipathogenic properties of PCL-Ag were tested against gram-negative and gram-positive bacteria, fungi, and biofilm formation. The results showed that the PCL-Ag nanofibers exhibit significant antimicrobial activity against a wide range of microorganisms, including those that cause human infections. The incorporation of Ag NPs into PCL nanofibers resulted in a self-sanitizing material that can be used in variety of applications, including wound dressings, water treatment, and air filtration. The development of a simple, scalable, and environmentally friendly method for the fabrication of these nanofibers is essential to ensure their widespread use in various industries. The ability to control the concentration and release rate of Ag ions in the PCL nanofibers will be critical to optimize their efficacy while minimizing their potential toxicity to human cells and the environment.

## 1. Introduction

The COVID-19 pandemic, caused by the rapid spread of the severe acute respiratory syndrome coronavirus (SARS-CoV-2), has led to the widespread use of facemasks as a preventive measure. According to reports from the World Health Organization (WHO), facemasks act as a barrier to prevent the transmission of infectious droplets and aerosols between individuals. However, recent studies have shown that widely used masks, including cotton masks, surgical masks, and N95 masks, do not provide 100% protective efficacy, but rather reduce the risk of contamination within a range of 20-90% when compared to not wearing a mask [1,2].

Surgical/N95 masks, which are more effective than cotton masks, provide 80-90% protection for a limited time (4 h), but are significantly less breathable and may cause discomfort during prolonged use. Furthermore, there is a risk of secondary infection due to the migration of pathogens in the filter layer [3], and frequent changes of masks can lead to problems of their utilization and environmental degradation [4].

Membranes based on electrospun biodegradable nanofibers have great potential for the development of facemask filters due to their nanosized pores, lightweight, enhanced air permeability, and high filtration efficiency [5,6]. In addition, inclusion therapeutic agents in the filter can prevent secondary contamination. Metal and metal oxide nanoparticles have shown a high potential as antipathogenic agents, since they have a wide spectrum of action and are able to inhibit viral, bacterial, and fungal infections [7,8,9].

One of the interesting approaches to incorporate metal ions is their capture by carboxyl groups through electrostatic interactions, which is commonly used in the capture of heavy metal ions and in coordination chemistry [10,11]. Homogeneous plasma polymers containing COOH groups can be deposited to create a large number of active sites [12], which lead to a uniform distribution of metal ions on the surface of PCL nanofibers [13]. The ease transition of Ag^+^ to Ag^0^ under UV irradiation makes the production of silver nanoparticles convenient [14].

Known for centuries, silver has antibacterial properties and is used for various medical purposes such as wound healing and water purification [15,16,17]. The antibacterial properties of silver are due to its ability to release silver ions (Ag^+^) in the presence of moisture or liquid. Silver ions can disrupt bacterial cell membranes and interfere with bacterial DNA replication and protein synthesis, leading to bacterial death. They can also inhibit enzymes involved in cellular respiration and energy production, further reducing bacterial survival [18,19]. The importance of silver ion release for antibacterial properties has led to the development of various Ag-based products, such as wound dressings, catheters, and implants. Silver modification helps prevent infections and improve patient outcomes, particularly in healthcare settings where bacterial infections are a major concern. Overall, the ability of silver ions to kill bacteria makes Ag-based products a valuable tool in fighting infections, and the release of silver ions is critical for their antibacterial properties [20].

Although the focus during the COVID-19 pandemic has been on fighting viral threats, multidrug-resistant bacteria present in the hospital environments pose a serious health problem [21]. Widespread antibiotic resistance poses a serious threat to global public health. Nanomaterial-based therapy is a promising strategy for the treatment of intractable bacterial and fungal infections [22].

Emergent fungal pathogens such as Candida are spreading in hospitals around the world, and the mortality rates for patients with invasive disease approach 60% [23]. Overall mortality from invasive candidiasis with *C. auris* ranges from 29% to 53%. *Candida parapsilosis* (*C. parapsilosis*) is a yeast that can be part of the healthy human microbiome but also causes invasive infection with a mortality rate of 20% to 45%. Its propensity to form biofilms makes it particular dangerous in central venous catheter infections [24]. *Candida albinos* (*C. albinos*) is a fungal pathogen that can be part of the healthy human microbiome, but can also cause mucosae infections or invasive candidiasis. One of the areas of application of nanomaterials in healthcare is their use for elimination of not only planktonic pathogen, but also for the prevention of biofilm formation [25]. *C. auris* produces biofilms, surface-adherent communities that resist antifungals and withstand desiccation [26]. Candida spp. proliferate as surface-associated biofilms in various clinical niches. These biofilms can be extremely difficult to eradicate in healthcare settings [27]. Thus, invasive candidiasis is a life-threatening disease with high mortality.

The objective of this research was to develop an environmentally friendly and scalable method to creating self-sanitizing electrospun nanofibers. This was achieved by immobilizing silver nanoparticles (Ag NPs) onto plasma-treated surfaces of biodegradable polycaprolactone (PCL) nanofibers. The plasma-deposited polymer layer containing carboxyl groups played a critical role in providing a uniform distribution of Ag NPs on the nanofiber surface. Ag ions were absorbed through electrostatic interaction and then reduced under UV-light. In this study, we demonstrated the high antibacterial and antifungal activity of the obtained PCL-Ag membranes, highlighting the potential for their use in facemasks due to their antipathogenic properties.

## 2. Materials and Methods

### 2.1. Electrospining of PCL

Nanofibers were obtained by electrospinning of a 9% PCL solution. To summarize, acetic acid (99%) and formic acid (98%) were employed to dissolve the granulated PCL. All materials were purchased from Sigma Aldrich (Darmstadt, Germany). The weight ratio of acetic acid (AA) to formic acid (FA) was 2:1. The samples were electrospun using a Super ES-2 machine manufactured by ESpin Nanotech (ESpin Nanotech, Kanpur, India), which contained both drum and static plate collectors. A static collection plate was employed in this study to collect the nanofibers. The flow rate of the PCL solution was 1 mL/h. The samples were collected onto polypropylene cloth at a distance of 12 cm from the nozzle. The electrospinining voltage was kept constant at 50 kV. The untreated, as-prepared PCL nanofibers are referred to as PCL-ref. More information on the electrospinning technique may be obtained elsewhere [28,29]. The PCL scaffold was 100 m thick.

### 2.2. Plasma COOH Coating and Deposition of Ag NPs on the Surfaces of Fibers

The COOH plasma polymer layers were deposited in a UVN-2M vacuum system with rotary- and oil-diffusion pumps. The plasma was ignited using an RFPG-128 disk generator (Beams & Plasmas, Moscow, Russia) connected to a radio frequency (RF) power source Cito 1310-ACNA-N37A-FF (Comet, Flamatt, Switzerland). The duty cycle and RF power were both set at 5% and 500 W. The reactor’s residual pressure was less than 103 Pa. The vacuum chamber was filled with CO_2_ (99.995%), Ar (99.998%), and C_2_H_4_ (99.95%) gases. A Multi-Gas Controller 647C (MKST, Newport, RI, USA) was used to control the gas flows. The Ar, CO_2_, and C_2_H_4_ flow rates were set to 50, 16.2, and 6.2 cm^3^/min, respectively. A VMB-14 unit (Tokamak Company, Dubna, Russia) and D395-90-000 BOC Edwards controllers were used to measure the pressure in the chamber. The RF electrode’s distance from the substrate was fixed at 8 cm. The deposition duration was 15 min, and the result was the formation of 100-nm-thick plasma coatings. Throughout this article, the plasma-coated PCL nanofibers are referred to as PCL-COOH. To immobilize Ag NPs on the surfaces of PCL-COOH fibers, we immersed PCL-COOH fibers in AgNO_3_ solution (C = 0.01 M) for 1 h, followed by irradiation and washing of prepared PCL Ag samples distilled water.

### 2.3. Characterization

Scanning-electron microscopy (SEM) was used to study the microstructures of as-prepared (PCL-ref) and surface-modified thin fibers (PCL-Ag and PCL-Ag-24 h) using a JSM-7600F Schottky field-emission scanning-electron microscope (JEOL Ltd., Singapore) equipped with an energy-dispersive X-ray (EDX) detector operating at 15 kV. To compensate for the surface charge and prevent sample damage, the samples were coated with a 40 nm thick Pt layer using a Smart Coater (JEOL Ltd.). We used EDX spectroscopy (EDXS) to examine the chemical and phase compositions using an 80-mm^2^ X-Max EDX detector (Oxford Instruments, Abingdon, UK). We carried out X-ray photoelectron spectroscopy (XPS) investigations using a PHI VersaProbe III spectrometer (ULVAC-PHI Inc., Osaka, Japan). The apparatus was fitted with a monochromatic Al K X-ray source (hv = 1486.6 eV), and investigations were performed at 23.5 eV pass energy and 50 W X-ray power. After removing the Shirley-type noise, the spectra were fitted using the CasaXPS program. The examined region had a maximum lateral resolution of 0.7 mm. By changing the hydrocarbon CHx component to 285.0 eV, the binding-energy scale was calibrated. Binding energies (BEs) were obtained from the literature for all carbon and oxygen environments [30,31,32].

### 2.4. Stability Test

The 15 × 15 mm PCL-Ag sample was submerged in 50 mL of distilled water for 24 h at room temperature. Next, the sample (PCL-Ag-24boh) was dried and analyzed using the previously mentioned SEM, EDXS, and XPS procedures.

### 2.5. Ag Ion Release

To determine the concentration of released silver, a 3535 mm PCL-Ag sample was submerged in 25 mL of PBS (pH 7.4). Aliquots of 5 mL were obtained after 3, 6, 24, and 72 h for spectroscopic examination using an inductively coupled plasma (ICP) atomic emission spectrometer (Agilent 4200 MP-AES, Santa Clara, CA, USA). For calibration, 1, 10, and 100 ppb silver standard solutions were prepared by diluting a silver standard solution (0.01 mg/mL) in 2% (*w*/*w*) HNO_3_. The results were normalized on 1 cm^2^ of PCL-Ag to correlate with results of antibacterial tests.

### 2.6. Antipathogen Activity

To analyze the antibacterial and antifungal activity of the samples PCL-ref and PCL-Ag, we used two types of hospital bacterial strain, Staphylococcus aureus CSA154 and Escherichia coli U20 (*S. aureus* CSA154, *E. coli* U20), and 3 types of fungi (*Candida albicans* ATCC90028, *Candida parapsilosis* ATCC90018 и *Candida auris* CBS10913) To study the antipathogen activity of planktonic forms, samples were immersed in pathogen solutions (normal saline (NS, NaCl = 9 g/L aqueous) containing 10^7^ CFU/mL (for the bacterial cultures) and 10^4^ CFU/mL (for the fungal cultures). After 6 and 24 h of incubation, the number of living planktonic pathogens was analyzed through decimal titration onto Petri dish plates with a dense nutrient medium for CFU counts. The prevention of biofilm formation was determined after sample incubation with bacterial/fungal cells at 37 °C for 24 h. The samples were removed from the plate well, gently washed three times to remove planktonic bacteria, and then sonicated on a Soniprep 150 homogenizer (MSE Ltd., Singapore). The treatment was carried out in 5 mL of 0.9% NaCl. The smaller the CFU, the better the anti-biofilm activity of the sample. The resulting suspensions with isolated microorganisms/fungi were used to determine the concentration of CFUs by the decimal dilution method in 0.3 mL of 0.9% NaCl. In total, 0.01 mL of the diluted bacterial/fungal suspension was placed on Petri dishes with the appropriate nutrient medium, dried in a closed dish at room temperature for 10 min, and then cultivated at 37 °C for 24 h. All procedures were carried out in a special room (box) in a laminar, in special clothes, by specifically trained personnel.

### 2.7. Biofilm Formation

The concentration of CFU was determined by decimal dilutions in 0.3 mL of saline. From each dilution, a 0.01 mL suspension was plated on Petri dishes with appropriate nutrient medium, dried in a closed dish at room temperature for 10 min, and cultured at 37 °C for 24 h.

## 3. Results

### 3.1. PCL-Ag Nanofibers’ Structures and Ag^+^ Release

The SEM micrographs of the PCL-ref and PCL-Ag nanofibers are presented in Figure 1. The average size of the nanofibers was 270 ± 50 nm. This average was estimated by measuring 100 fibers that were randomly selected from each sample using ImageJ software. After modifications, the nanofibers were not damaged, but the surface of the PCL-Ag was rougher than that of the PCL-ref. The EDXS analysis indicated that the Ag were uniformly distributed over the fiber surface of the PCL-Ag, although large agglomerates of Ag NPs were also detected. The results of the determination of the chemical compositions of the samples by the EDXS before and after the stability test are summarized in Table 1. According to Table 1, soaking in water for 24 h reduced the amount of Ag NPs in the composition fibers by 0.7 at.%.

To characterize the surface compositions of the samples, we employed XPS analysis. The atomic compositions of all the samples are reported in Table 2. The high level of loading with Ag nanoparticles was previously confirmed by the significant concentration (5 at.%) of Ag. The amount of silver significantly changed after soaking in the deionized water for 24 h; however, a significant portion of Ag NPs remained on the surface, confirming the formation of bonds between the Ag nanoparticles and the nanofibrous matrix. The material compositions differed from the EDXS results (Table 1), probably due to the different depths of analysis: ~10 nm (XPS) and ~1000 nm (EDXS). The significant difference in concentration results means that only the thin surface layers contained Ag NPs. Moreover, it seems that some of the Ag NPs were not bonded with carboxyl (COOH) groups, but were simply fixed in the 3D structures of the fibers,; thus, after immersion in water, unbonded NPs were released. This suggestion was also confirmed by the ICP AES results, according to which the rapid release of Ag ions was detected in the first 3 h, followed by a slow kinetic release of Ag ions (Figure 2).

The XPS analyses revealed significant differences between the compositions of the samples due to the different probing depths. As shown in Table 2, the concentration of silver was the highest, while the concentration of carbon was the lowest. Furthermore, some nitrogen was detected. In order to reveal the environment with all the elements, we performed a detailed analysis of the XPS peaks.

To further determine the environment of the surfaces of the PCL nanofiber, including the Ag nanoparticles, we analyzed the C1s, O1s, N1s, and Ag3d spectra. The XPS C1s spectrum of the PCL-ref (Figure 3A), was fitted with a sum of three components, namely hydrocarbons CHx (BE = 285 eV), an ether group C-O (BE = 286.4 eV), and an ester group C(O)O (BE = 289.0 eV), as reported elsewhere [33,34]. The XPS C1 signal of the PCL-Ag (Figure 3B) was fitted using four components, CHx, C-O, C=O, and C(O)O, and the CHx was the dominant environment.

Further interesting results were obtained from the O1s spectra depicted in Figure 3E,F. The O1s spectrum of the PCL-Ag nanoparticles was mainly composed of carbon–oxygen bonds (C-O and C=O, centered at 532.1 eV and 533.3 eV, respectively, FWHM = 1.7 eV), but we also detected an Ag_2_O bond, located at 530.8 eV (FWHM = 1.7 eV), a typical position for oxygen in metal oxides. The silver nanoparticles covered the surfaces of the fibers, and 3.5% Ag was detected by XPS (Table 1). The XPS Ag3d5/2 signal was fitted by one component: either metallic silver or silver oxide (I) Ag^0^/Ag^+^ (BE = 368.0 ± 0.1 eV, FWHM = 1.2 eV). After soaking in water, the concentration of silver slightly decreased; nevertheless, 2.4 at% of silver can be considered as very high concentration.

The N1s environment before and after immersion in water is shown in Figure 3I,J. The spectrum before the immersion was fitted with three components: amides N-C=O (BE = 399.8 eV), protonated amines NH_3_^+^ (BE = 401.6 eV) and nitrates NO_3_^−^ (BE = 406.4 eV). After soaking in water for 24 h, practically only N-C=O was visible, with a minor concentration of protonated amines; no nitrates were detected. The presence of nitrate NO_3_- at a rate of 0.6 at.% indicates that some unreacted silver nitrate was present on the surface. Its release during washing can explain the decay of the silver concentration after the immersion in water. This also had an effect on the burst silver ions released in the first hours.

### 3.2. Antipathogen Activity

The results of this study indicate that the PCL-Ag samples exhibited potent antibacterial effects against all the tested strains, as shown in Figure 4 and Figure 5. After 6 h of incubation, we observed 100% antibacterial activity; no values of *E. coli* U20 bacteria (as shown in Figure 4A) or *Candida auris* CBS10913 (as shown in Figure 5A) were detected. The other strains, including *Candida parapsilosis* ATCC90018, *Candida albicans* ATCC90028, and *S. aureus* MW2, showed decreases in colony-forming units (CFUs), ranging from 2 to 4 log (as depicted in Figure 5B, Figure 5C and Figure 4B, respectively).

In contrast, all the unmodified tissue samples exhibited varying densities of biofilm. However, none of the Ag-modified tissue samples showed any biofilm formation, as shown in Figure 4C and Figure 5D. These findings indicate that the PCL-Ag nanofibers were effective in preventing biofilm formation, which is critical for preventing infections in medical applications.

Hence, our results demonstrate the significant antimicrobial activity of PCL-Ag nanofibers against a broad spectrum of microorganisms.

## 4. Discussion

In this work, we have shown that our robust technique ensures high levels of Ag^+^ loading and its fast and long-term release. Our results demonstrate significant enhancement compared with other methods. Generally, the measurement of silver loadings in nanofibers to determine antibacterial effects can vary, depending on the preparation method and the type of polymer used. Studies reported silver-loading concentrations ranging from 0.05% to 8.0% (*w*/*w*) in nanofibers, with higher concentrations leading to more significant antibacterial effects. However, it is important to note that the optimal silver-loading concentration may vary, depending on the specific application and the target pathogen.

Dubey et al. [35] described a new approach to the synthesis of Ag NPs with average sizes of 15–20 nm based on the mixture of high-molecular-weight polyethylene oxide (PEO) with AgNO_3_ solution, followed by the formation of electrospun nanofibers PCL:PEO (1:4 wt.%), with the incorporation of 1–3 wt.% Ag NPs, and tested their antibacterial activity. They demonstrated that the incorporation of Ag NPs into blended PCL:PEO nanofibers led to decreases in the fiber diameters to 70–150 nm vs. 150–300 nm for the pure PCL:PEO fibers, which can be explained by the changing conductivity of the formation solution. The same effect of decreasing diameters of NFs with increasing concentrations of Ag was detected by Annur et al. in composite fibers (CS:PEO:Ag NPs) [X]. All the composite samples (PCL:PEO/Ag NPs) in that study demonstrated high levels of antimicrobial activity, suppressing the growth of antibiotic-resistant GFP *E. coli* bacteria. The authors detected the rapid release Ag ions in the first 20 h (40% of loading), followed by a slow increase in the concentration of Ag in the next 76 h.

Annur et al. [36] used Ar plasma to obtain chitosan (CS):PEO:Ag NP composite nanofibers. In the first step, the nanofibers were produced from a mixture of CS:PEO:AgNO_3_ (5.4:0.6, with the AgNO_3_ concentrations varying from 0.5 to 2 wt%, respectively). A plasma treatment was carried out using a RF power of 200 W, a chamber pressure of 200 mTorr, and an argon flow of 32 cm^3^/min. The optimum time for the plasma treatment was determined to be 1.5 min. To increase the stability of the samples, after the plasma treatment, they were immersed in an aqueous TPP solution. The average size of the Ag NPs was determined to be 1.5 nm. The addition of Ag increased the antimicrobial properties of prepared materials against *E. coli* bacteria by seven times.

Foroushani et al. [37] synthesized silk–chitosan nanofibers loaded with curcumin and Ag NPs for wound-healing applications. They demonstrated that functional groups of polymers (amino groups of chitosan and hydroxyl groups of curcumin) bonded Ag ions, forming a polymeric network with a uniform distribution of silver. However, the excellent antibacterial properties of the synthesized material against *E. coli* and *S. aureus* bacteria were represented by the activities of Ag NPs, curcumin, and chitosan.

The incorporation of silver in the form of nanoparticles (AgNPs) into PCL nanofibers enhances their antibacterial properties. One of the key advantages of using PCL nanofibers with AgNPs is their ability to release silver ions very rapidly [38,39,40]. The fast release of silver ions from PCL nanofibers is due to several factors. Firstly, PCL is a hydrophobic polymer, which limits the interaction between the material and the water molecules. However, the incorporation of AgNPs into PCL nanofibers can improve their hydrophilicity, leading to better water absorption and the faster release of silver ions.

Secondly, the small size of AgNPs in the range of 5–50 nm facilitates their diffusion through the PCL-nanofiber matrix, allowing the more rapid release of silver ions [41,42]. The high surface-area-to-volume ratio of the nanoparticles also increases their reactivity, leading to the more rapid release of silver ions upon contact with fluids [34].

Finally, the release of silver ions from PCL nanofibers with AgNPs can be affected by the concentration of silver and the morphology of the nanofibers. Higher concentrations of silver and nanofibers with smaller diameters can lead to the more rapid release of silver ions.

The swift release of silver ions from PCL nanofibers with AgNPs is desirable for wound-dressing applications as it can provide a rapid antibacterial effect, preventing the growth and spread of bacteria at the wound site. It is worth noting that the rapid release of silver ions may also increase the risk of toxicity to human cells and, therefore, it is important to find an optimal silver-loading concentration and release rate that provides effective antibacterial properties without harming human cells [36].

The aim of this study was to create an eco-friendly and scalable method to produce self-sanitizing electrospun nanofibers. The Ag NPs were immobilized on the surfaces of plasma-modified biodegradable PCL nanofibers. The plasma-polymerized layer deposited from the Ar/CO_2_/C_2_H_4_ gas mixture contained carboxyl groups that played a crucial role in ensuring the even distribution of the Ag NPs on the nanofiber surface. Electrostatic interactions with negatively changed COOH groups allowed the Ag ions to be absorbed and reduced under UV light.

Here, we investigated the effectiveness of PCL-Ag electrospun nanofibers against different pathogenic strains, including *E. coli* U20, *Candida auris* CBS10913, *Candida parapsilosis* ATCC90018, *Candida albicans* ATCC90028, and *S. aureus* MW2. We found that the PCL-Ag samples demonstrated potent antibacterial activity against all the tested strains.

After six hours of incubation, 100% antipathogen activity was observed for the E.coli U20 and *Candida auris* CBS10913. This means that no colonies of these pathogens were found, indicating the complete inhibition of their growth. The other strains, including *Candida parapsilosis* ATCC90018, *Candida albicans* ATCC90028, and *S. aureus* MW2, also showed a significant reduction in the number of colonies formed, ranging from 2 to 4 log scales.

In contrast, the unmodified tissue samples showed biofilm formation with varying densities. Biofilms are common defense mechanisms used by pathogens to protect themselves from the host’s immune system and antimicrobial agents. The biofilms formed on the unmodified tissue samples may have made them more resistant to antibiotics and antimicrobial agents, making infections more difficult to treat.

It is worth noting that none of the Ag-modified tissue samples showed biofilm formation. The incorporation of silver nanoparticles into the PCL-Ag electrospun nanofibers may have contributed to the prevention of biofilm formation. Silver nanoparticles have been shown to have antimicrobial properties and can inhibit biofilm formation by disrupting bacterial and fungal cell membranes. Therefore, PCL-Ag electrospun nanofibers can effectively inhibit the growth of various pathogenic strains, including those that are resistant to antibiotics. The incorporation of silver nanoparticles into nanofibers may provide additional antimicrobial properties, preventing biofilm formation and potentially reducing the risk of infections.

The high levels of antibacterial and antifungal activity of PCL-Ag membranes can be attributed to several factors: the presence of Ag nanoparticles on the surfaces of the nanofibers, which act as potent antipathogenic agents; some silver nitrates, as detected by XPS; and 0.6 at.% of nitrogen from the nitrate environment present on the surface.

During washing or immersion in water, the unreacted silver nitrate could be released, resulting in the decay of the silver concentration. This means that the silver ions that were initially present on the surfaces of the nanofibers may have been released into the water. The release of silver ions can have both positive and negative effects. On one hand, it can lead to burst-silver-ion release in the first hour, which can increase the antibacterial and antifungal efficacy of PCL-Ag electrospun nanofibers. This means that a high concentration of silver ions is released initially, which can provide an immediate antimicrobial effect. However, it should be remembered that on the other hand, the release of silver ions can also have negative effects. Silver ions are toxic to both bacteria and human cells, and their release can cause damage to healthy cells. Therefore, the amount and rate of silver ions released from PCL-Ag electrospun nanofibers need to be carefully controlled to balance the desired antimicrobial effect with the potential negative effects.

Overall, the presence of nitrate NO_3_^-^ in the PCL-Ag electrospun nanofibers indicates the presence of unreacted silver nitrate on the surface, which can be released during washing or immersion in water. This release of silver ions can have both positive and negative effects, and careful control of the release rate is necessary to maximize the antibacterial and antifungal efficacy of nanofibers while minimizing potential negative effects.

Finally, the results of the study confirm that PCL-Ag nanofibers have the potential to be utilized as effective antipathogenic materials in the development of face masks and other medical devices, as well as in wound dressings and other biomedical applications. The lack of biofilm formation in Ag-modified tissue samples is particularly relevant to hospital environments, where multidrug-resistant bacteria are significant sources of health problems. The use of PCL-Ag nanofibers could provide a solution to this problem by inhibiting the formation of biofilms and providing sterile environments for patients. Further studies can be conducted to optimize the Ag loading and the release rate of PCL-Ag nanofibers to improve their overall antibacterial efficacy and to explore their potential for clinical applications.

## 5. Conclusions

In the present study we successfully developed a scalable and environmentally friendly method to fabricate self-sanitizing electrospun nanofibers. The immobilization of silver nanoparticles (Ag NPs) onto the surfaces of plasma-treated biodegradable polycaprolactone (PCL) nanofibers, through electrostatic interaction and UV-light reduction, resulted in a material with significant antimicrobial activity against a wide range of microorganisms, including those that cause infections in humans. The plasma-deposited polymer layer containing carboxyl groups played a critical role in ensuring the even distribution of Ag NPs on the nanofiber surface. A high concentration of Ag+ ions was released within the first three hours of the immersion in water, but this ion release subsequently decreased and stabilized. The incorporation of the Ag NPs into the PCL nanofibers resulted in a self-sanitizing material that can be used in various applications, including wound dressings, water treatment, and air filtration. The ability to control the concentration and release rate of Ag NPs in the PCL nanofibers will be critical to optimize their efficacy while minimizing their potential toxicity to human cells and the environment. Overall, the results suggest that PCL-Ag nanofibers have significant potential for use in various biomedical and environmental applications.

## Figures and Tables

**Figure 1 jfb-14-00336-f001:**
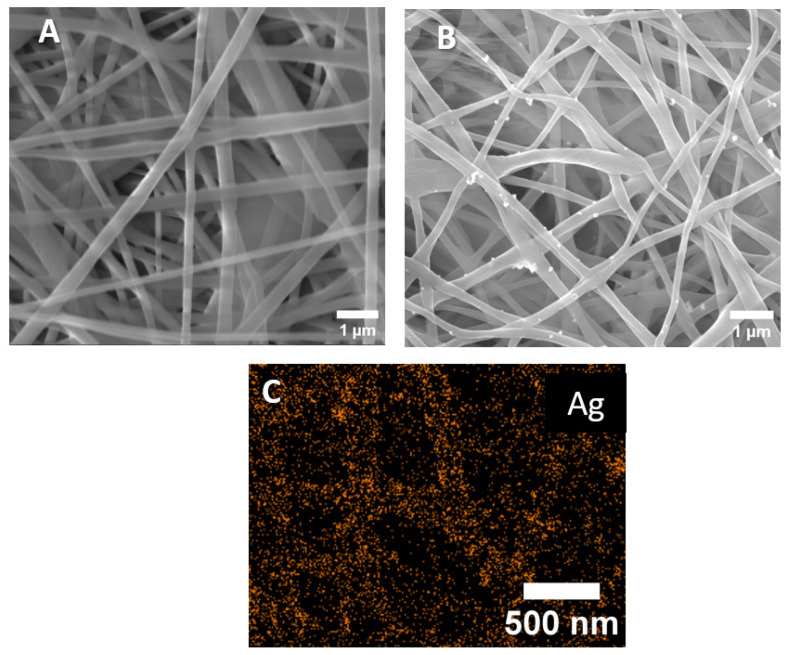
SEM images of PCL-ref (**A**), PCL-Ag (**B**), and EDXS elemental map of Ag on the surface PCL-Ag (**C**).

**Figure 2 jfb-14-00336-f002:**
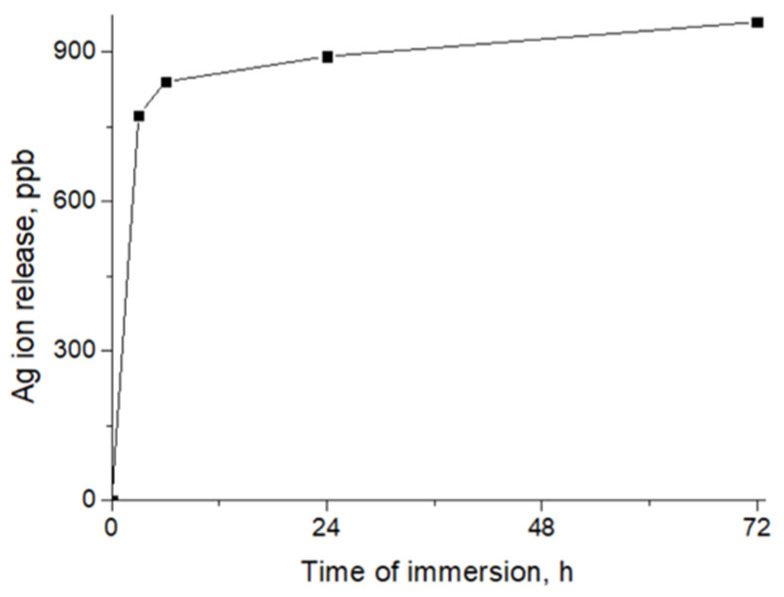
Release of Ag ions from PCL-Ag, as measured by ICP AES.

**Figure 3 jfb-14-00336-f003:**
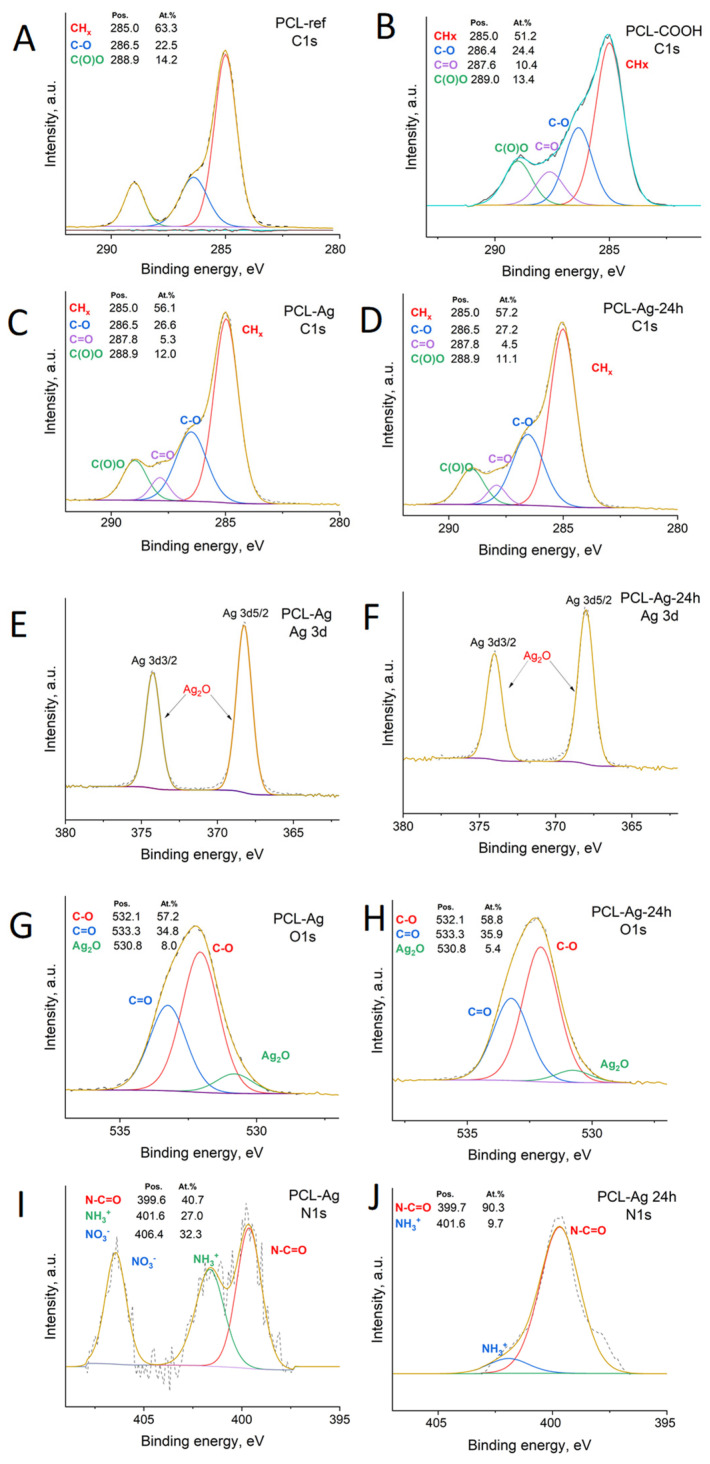
XPS C1s PCL-ref(**A**), PCL-COOH (**B**), PCL-Ag (**C**), and PCL-Ag-24 h (**D**). XPS Ag3d (**E**,**F**), O1s (**G**,**H**), and N1s (**I**,**J**) spectra of PCL-Ag and PCL-Ag-24 h after water treatment.

**Figure 4 jfb-14-00336-f004:**
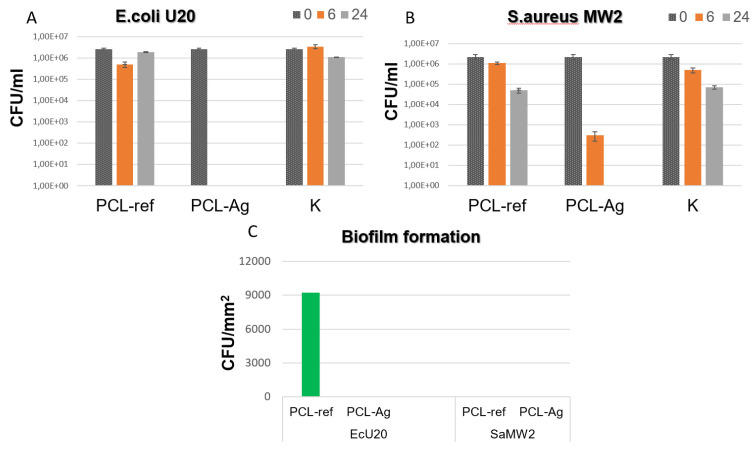
CFU concentration of Escherichia coli U20 (**A**) and Staphylococcus aureus MW2 (**B**) strains after incubation with unmodified (ref) and modified (Ag) samples and in control sample without tissue (K). CFU density of biofilms (**C**) of Escherichia coli U20 and Staphylococcus aureus MW2 strains on unmodified (ref) and modified (Ag) samples.

**Figure 5 jfb-14-00336-f005:**
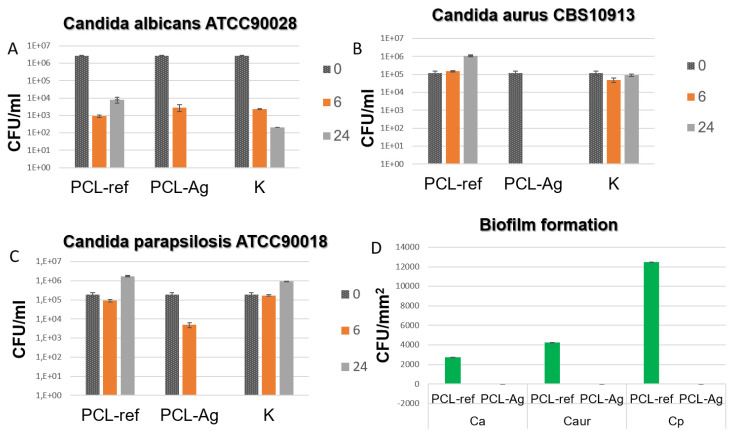
CFU concentration of different fungal strains (**A**–**C**) after incubation with unmodified (ref) and modified (Ag) samples and in control sample without tissue (K). CFU density of biofilms (**D**) of Escherichia coli U20 and Staphylococcus aureus MW2 strains on unmodified (ref) and modified (Ag) samples.

**Table 1 jfb-14-00336-t001:** Atomic compositions of the samples measured by EDXS.

Sample	[C], at.%	[O], at.%	[Ag], at.%	[Pt], at.%
PCL-ref	73.9	26.0		0.1
PCL-Ag	89.9	9.9	1.1	0.1
PCL-Ag-24 h	91.1	8.4	0.4	0.1

**Table 2 jfb-14-00336-t002:** Atomic compositions of the samples, as measured by XPS.

Sample	C1s, at.%	O1s, at.%	Ag3d, at.%	N1s, at.%
PCL-ref	74.0	26.0	-	-
PCL-COOH	72.5	27.5	-	-
PCL-Ag	67.7	26.7	3.5	2.1
PCL-Ag-24 h	71.9	23.9	2.3	1.9

## Data Availability

The data is available from corresponding authors upon a reasonable request.

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
