# Peer review of "Self-Sanitizing Polycaprolactone Electrospun Nanofiber Membrane with Ag Nanoparticles"

_jfb, 2023, doi:10.3390/jfb14070336_

Round 1
Reviewer 1 Report
The work deals with results of a simple, environment-friendly method for self-sanitizing electrospun nanofibers production. Authors immobilized silver nanoparticles onto surface plasma-treated biodegradable polycaprolactone nanofibers. After, the particles were absorbed through electrostatic interaction and reduced under UV-light. Tests showed that nanofibers exhibited significant antimicrobial activity against various microorganisms. Authors determined the average size of the nanofibers by SEM measurements, and investigated the surface compositions of the samples through XPS analysis, where significantly different compositions of samples under different probing depths were verified. The general conclusion is that the incorporation of silver in the form of nanoparticles into polycaprolactone nanofibers enhances their antibacterial properties. The work has interest and important information, but before acceptance, I think some modifications in the form are necessary.
1. In the introduction, lines 108 - 117, all text must be cut off: "introduction should briefly place the study in a broad context and highlight why it is important. It should define the purpose of the work and its significance. The current state of the research field should be carefully reviewed and key publications cited. Please highlight controversial and diverging hypotheses when necessary. Finally, briefly mention the main aim of the work and highlight the principal conclusions. As far as possible, please keep the introduction comprehensible to scientists outside your particular field of research. References should be numbered in order of appearance and indicated by a numeral or numerals in square brackets—e.g., [1] or [2, 3], or [4, 5, 6]. See the end of the document for further details on references."
2. Please, cut off the sentences in the lines 433- 438: "Data Availability Statement: We encourage all authors of articles published in MDPI journals to share their research data. In this section, please provide details regarding where data supporting reported results can be found, including links to publicly archived datasets analyzed or generated during the study. Where no new data were created, or where data is unavailable due to privacy or ethical restrictions, a statement is still required. Suggested Data Availability Statements are available in section “MDPI Research Data Policies” at https://www.mdpi.com/ethics."
3. Part of the discussion (lines 330-343) repeats sentences of the introduction (lines 100-107, i.e., "The objective of this research was (...) in face masks for their antipathogenic properties." Please, improve this part of the discussion.
4. The inset of Fig. 1(B) is not easy to understand. Please, improve it.
5. In Fig. 3 (A-J), if it is an arbitrary scale, then authors must cut off the numbers in the vertical scales. Also, please add "Intensity, a.u." in Fig. 3(J).
In general the English language is correct, only small changes are necessary;
ex: "Our work has demonstrated..." instead of "Our work has been demonstrated..." etc.
Author Response
First of all, we would like to thank the Reviewer for his/her great effort that allowed us to significantly improve our manuscript. We did comprehensive revision added many new references, reworked figures and changed text accordingly. All our modifications are highlighted.
Q1. In the introduction, lines 108 - 117, all text must be cut off: "introduction should briefly place the study in a broad context and highlight why it is important. It should define the purpose of the work and its significance. The current state of the research field should be carefully reviewed and key publications cited. Please highlight controversial and diverging hypotheses when necessary. Finally, briefly mention the main aim of the work and highlight the principal conclusions. As far as possible, please keep the introduction comprehensible to scientists outside your particular field of research. References should be numbered in order of appearance and indicated by a numeral or numerals in square brackets—e.g., [1] or [2, 3], or [4, 5, 6]. See the end of the document for further details on references."
Author Response: The paragraph was deleted
Q2. Please, cut off the sentences in the lines 433- 438: "Data Availability Statement: We encourage all authors of articles published in MDPI journals to share their research data. In this section, please provide details regarding where data supporting reported results can be found, including links to publicly archived datasets analyzed or generated during the study. Where no new data were created, or where data is unavailable due to privacy or ethical restrictions, a statement is still required. Suggested Data Availability Statements are available in section “MDPI Research Data Policies” at https://www.mdpi.com/ethics."
Author Response: The paragraph was deleted.
Q3. Part of the discussion (lines 330-343) repeats sentences of the introduction (lines 100-107, i.e., "The objective of this research was (...) in face masks for their antipathogenic properties." Please, improve this part of the discussion.
Author Response: The discussion part was rewritten.
Q4. The inset of Fig. 1(B) is not easy to understand. Please, improve it.
Author Response: The Fig 1 was improved
Q5. In Fig. 3 (A-J), if it is an arbitrary scale, then authors must cut off the numbers in the vertical scales. Also, please add "Intensity, a.u." in Fig. 3(J).
Author Response: The Fig 3 was improved according recommendation
Reviewer 2 Report
The manuscript, while dealing with a topic of interest for the journal's readership and reporting interesting results, requires considerable revision work.
Data are not clearly and fully described in the results section and the Discussion section contains a lot of repetition.
Some specific points to be addressed:
- the first sentences of the abstract are identically repeated at the end of the introduction and the full abstract is reported at lines 330-342 of the Discussion section).
- The authors forgot to remove journal indications (see lines 108-117, or 407-410)
- In section 2.6, line 196, 2h or 6h and 24h as in Figure 4?
- In this reviewer opinion the approach to detect biofilm formation is not fully safe. The material should be gently washed before being immersed in NS, to remove planktonic bacteria. Saline solution alone does not always assure a complete removal of biofilm, a sonication in an ultrasound bath should be done.
The quality of English language makes some sections of the manuscript very difficult to understand
Author Response
Reviewer 2
First of all, we would like to thank the Reviewer for his/her great effort that allowed us to significantly improve our manuscript. We did comprehensive revision added many new references, reworked figures and changed text accordingly. All our modifications are highlighted.
The manuscript, while dealing with a topic of interest for the journal's readership and reporting interesting results, requires considerable revision work.
Q1 Data are not clearly and fully described in the results section and the Discussion section contains a lot of repetition.
Author Response: The text was modified accordingly all changes were highlighted
Q2 the first sentences of the abstract are identically repeated at the end of the introduction and the full abstract is reported at lines 330-342 of the Discussion section).
Author Response: We have modified the Discussion section significantly. The repletion was corrected.
Q3 The authors forgot to remove journal indications (see lines 108-117, or 407-410)
Author Response: The journal indications were removed
Q4 In section 2.6, line 196, 2h or 6h and 24h as in Figure 4?
Author Response: We made two experiments to research antipathogenic activity of samples. First part of experiments is to study inhibition of pathogen growth in solutions For that purpose we immersed samples into infectious solutions contained of 107 CFU/mL (for the bacterial cultures) and 104 CFU/ml (for the fungal culture). and after 6 and 24 h we analyzed content of pathogen. The second part of experiments is to analyze antipathogenic activity against biofilm formation, infectious suspension was inoculated onto Petri dishes with appropriate nutrient media, dried in a sealed cup at room temperature for 10 min, and the cultures were cultured at 37 °C for 24 h. On the next step, samples were placed in a contaminated petri dish and incubated for 2 h, followed gently washing of material and determination CFU amount.
Q5 In this reviewer opinion the approach to detect biofilm formation is not fully safe. The material should be gently washed before being immersed in NS, to remove planktonic bacteria. Saline solution alone does not always assure a complete removal of biofilm, a sonication in an ultrasound bath should be done.
Author Response: The prevention of biofilm formation was determined after sample incubation with bacterial/fungal cells at 37°C for 24 h. The samples were removed from the plate well, gently washed three times to remove planktonic bacteria, and then sonicated on a Soniprep 150
homogenizer (MSE Ltd). The treatment was carried out in 5 mL of 0.9% NaCl. All procedures were carried out in a special room (box) in a laminar in special clothes and by specially trained personnel.
Reviewer 3 Report
Review “Self-sanitizing Polycaprolactone Electrospun Nanofiber Mem-2 brane with Ag Nanoparticles”
Silver particles were attached on plasma activated surfaces of PCL nanofibers. After an initial burst there is prolonged release of Ag from the sample and antibacterial properties. Ag particles are evenly distributed on the fibers.
Line 25: “sufficient amount of” – Could this amount be quantified?
Line 43: “infected” – “infectious”
Line 52: “utilization” – It is not clear what is meant here. How is utilization related to frequently changing the masks?
Line 60: “incorporating” – “incorporate”
Line 62: “delaying heavy metal ions” – I don’t understand this example. Delay for what?
Line 62-66: References are needed.
Line 70-79: References are needed.
Line 98: Extra “)”
Line 108-117: “introduction should 108 briefly place the study in a broad context and highlight why it is important. It should 109 define the purpose of the work and its significance. The current state of the research field 110 should be carefully reviewed and key publications cited. Please highlight controversial 111 and diverging hypotheses when necessary. Finally, briefly mention the main aim of the 112 work and highlight the principal conclusions. As far as possible, please keep the introduc-113 tion comprehensible to scientists outside your particular field of research. References 114 should be numbered in order of appearance and indicated by a numeral or numerals in 115 square brackets—e.g., [1] or [2, 3], or [4, 5, 6]. See the end of the document for further 116 details on references.” I think these are comments which should be deleted.
Line 129: Please remove “and placed”
Line 122 and line 131 have similar sentences and should be combined: “Further information about the electrospinning 131 process can be found elsewhere [23].”
Line 130 and line 132: Information is repeated “The authors used the term PCL-ref to refer to the untreated, as-prepared PCL nanofibers.” And “Spinning PCL samples were denoted as PCL-ref.”
Line 133: How long was the electrospinning conducted or how much solution was electrospun?
Line 140: A bracket is missing “)”
Line 142: pressure number has a problem. Maybe better 1mPa.
Line 145 and 151, 184, 244: “C2H4” and “AgNO3”, “HNO3“, “Ag2O” not correctly shown
Line 145: The unit of flowrate “sccm” is not SI unit and I advise to convert to cm3/s or write it as cm3/min.
Line 152: Is there a special name for the Ag-coated samples? In line 155 is PCL-Ag and PCL-Ag-24h are used). Is the name PCL-COOH necessary?
Line 158-160: Is Pt coating necessary for PCL-Ag since the surface is conductive with Ag coating? For PCL I would not coat with Pt.
Line 161: “80-mm2” Is this mm2? Is EDX analysis possible for Pt coating? Line 174: Please remove the space in the sample name to be the same as before “PCL-Ag-24*h”
Line 179: “boron”? Not Ag?
Line 184: “recalculated” could be replace by “normalized”
Line 190: add “we” before “used”
Line 194: “107 CFU/mL” or “107 CFU/mL”
Line 211: “but surface PCL-Ag is smoother as compared to PCL - ref” How is this seen? PCL-Ag seems to have particles on the surface which is increasing surface roughness.
Line 215: remove “How”
Table 1 and 2: is there a unit for the numbers or is it [%]?
Line 216: “from the composition fibers at 0.7 at.%.” should be changed to “from the composition fibers by 0.7 at.%.”
Line 221: “confirming the formation of bonding”
Line 224: “The significance differ” to “The significant difference”
Line 227: “unbonded NPs were delayed.” To “unbonded NPs were released”
Review English language. Some sentence need to be shortened.
Line 238: “27,28”
Line 243: unit missing for “533.3”
Line 261: “After 6 hours of incubation, we observed a 100% reduction in the number of E.coli U20 bacteria (as shown in Fig. 4a)” For E.coli is no value at 6h.
Figure 4/5: image and caption uses “ref” instead of “PCL-ref” and “Ag” instead of “PCL-Ag”
Figure 4c/5A has no data for PCL-Ag.
Figures 4/5: Is this for 6hours? The colors of A and C should match.
Line 267-271: What is the limit of CFU for biofilm formation?
Line 303-309: More qualitative comparison with literature is needed. Better than which state-of-the-art? ref e.g. “Silver-nanoparticle-Incorporated composite nanofibers for potential wound-dressing applications”, Poornima Dubey, Bharat Bhushan, Abhay Sachdev, Ishita Matai, S. Uday Kumar, P. Gopinath, First published: 06 June 2015 https://doi.org/10.1002/app.42473 has similar release characteristics and antibacterial activity.
Line 321-323: Reference?
Line 330: “simple, eco-friendly, and easily scalable method” Is the method of plasma coating really more simple than one-pot electrospinning and how is it scalable?
Line 333:, 417/418 “The polymer layer deposited through” I think plasma treatment activates COOH in a surface layer of PCL but there is no deposition of polymer on the polymeric electrospun PCL fibers.
none
Author Response
First of all, we would like to thank the Reviewer for his/her great effort that allowed us to significantly improve our manuscript. We did comprehensive revision added many new references, reworked figures and changed text accordingly. All our modifications are highlighted.
Silver particles were attached on plasma activated surfaces of PCL nanofibers. After an initial burst there is prolonged release of Ag from the sample and antibacterial properties. Ag particles are evenly distributed on the fibers.
· Line 25: “sufficient amount of” – Could this amount be quantified?
Author Response: The text was corrected as follows: “a sufficient amount of silver nanoparticles on the surface (⁓2.3 vs 3.5 at.% were determined by XPS analysis)”
Line 43: “infected” – “infectious”
Author Response: Done
Line 52: “utilization” – It is not clear what is meant here. How is utilization related to frequently changing the masks?
Author Response: Frequent mask changes (recommendations for surgical masks every 3-4 hours) leads to increased recycling of used masks, and since they consist of polypropylene, which tends to form microplastics, this can lead to environmental problems
Line 60: “incorporating” – “incorporate”
Author Response: Done
Line 62: “delaying heavy metal ions” – I don’t understand this example. Delay for what?
Author Response: We are apologize, “delaying” was changed on “capturing”
Line 62-66: References are needed.
Author Response: The next references were added: “Homogeneous plasma polymer containing COOH groups can be deposited to create a large number of active sites [12] that lead to the uniform distribution of metal ions on the surface of PCL nanofibers.[13] The ease of transitioning Ag+ to Ag0 under UV irradiation makes the production of silver nanoparticles convenient. [14]”
- Permyakova, E.S.; Polčak, J.; Slukin, P.V.; Ignatov, S.G.; Gloushankova, N.A.; Zajíčková, L.; Shtansky, D.V.; Manakhov, A. Antibacterial Biocompatible PCL Nanofibers Modified by COOH-Anhydride Plasma Polymers and Gentamicin Immobiliza-tion. Mater. Des. 2018, 153, doi:10.1016/j.matdes.2018.05.002.
- Nowak, S.; Mauron, R.; Dietler, G.; Schlapbach, L. XPS-Study of Metal-Polymer Interfaces After Polymer Surface Treatment by Ion and Plasma Techniques BT - Metallized Plastics 2: Fundamental and Applied Aspects. In; Mittal, K.L., Ed.; Springer US: Boston, MA, 1991; pp. 233–244 ISBN 978-1-4899-0735-6.
- Wang, H.; Zhang, G.; Mia, R.; Wang, W.; Xie, L.; Lü, S.; Mahmud, S.; Liu, H. Bioreduction (Ag+ to Ag0) and Stabilization of Silver Nanocatalyst Using Hyaluronate Biopolymer for Azo-Contaminated Wastewater Treatment. J. Alloys Compd. 2022, 894, 162502, doi:https://doi.org/10.1016/j.jallcom.2021.162502.
Line 70-79: References are needed.
Author Response: The next references were added
- Gibała, A.; Żeliszewska, P.; Gosiewski, T.; Krawczyk, A.; Duraczyńska, D.; Szaleniec, J.; Szaleniec, M.; Oćwieja, M. Antibacterial and Antifungal Properties of Silver Nanoparticles—Effect of a Surface-Stabilizing Agent. Biomolecules 2021, 11, 1–20, doi:10.3390/biom11101481.
- Lansdown, A.B.G. Silver. I: Its Antibacterial Properties and Mechanism of Action. J. Wound Care 2002, 11, 125–130, doi:10.12968/jowc.2002.11.4.26389.
- Kędziora, A.; Wieczorek, R.; Speruda, M.; Matolínová, I.; Goszczyński, T.M.; Litwin, I.; Matolín, V.; Bugla-Płoskońska, G. Comparison of Antibacterial Mode of Action of Silver Ions and Silver Nanoformulations With Different Physico-Chemical Properties: Experimental and Computational Studies. Front. Microbiol. 2021, 12, 1–12, doi:10.3389/fmicb.2021.659614.
- Line 98: Extra “)”
Author Response: the extra symbol “)” was deleted
Line 108-117: “introduction should 108 briefly place the study in a broad context and highlight why it is important. It should 109 define the purpose of the work and its significance. The current state of the research field 110 should be carefully reviewed and key publications cited. Please highlight controversial 111 and diverging hypotheses when necessary. Finally, briefly mention the main aim of the 112 work and highlight the principal conclusions. As far as possible, please keep the introduc-113 tion comprehensible to scientists outside your particular field of research. References 114 should be numbered in order of appearance and indicated by a numeral or numerals in 115 square brackets—e.g., [1] or [2, 3], or [4, 5, 6]. See the end of the document for further 116 details on references.” I think these are comments which should be deleted.
Author Response: The paragraph was deleted
Line 129: Please remove “and placed”
Author Response: “and placed’” was removed
Line 122 and line 131 have similar sentences and should be combined: “Further information about the electrospinning 131 process can be found elsewhere [23].”
Author Response: the sentences were combined
Line 130 and line 132: Information is repeated “The authors used the term PCL-ref to refer to the untreated, as-prepared PCL nanofibers.” And “Spinning PCL samples were denoted as PCL-ref.”
Author Response: the repeated information was deleted “Spinning PCL samples were denoted as PCL-ref.”
Line 133: How long was the electrospinning conducted or how much solution was electrospun?
Author Response: We used 2 ml polymer solution to produce sample 20x20 cm.
Line 140: A bracket is missing “)”
Author Response: the bracket was added
Line 142: pressure number has a problem. Maybe better 1mPa.
Author Response: The pressure number was corrected
Line 145 and 151, 184, 244: “C2H4” and “AgNO3”, “HNO3“, “Ag2O” not correctly shown
Author Response: The subscripts were corrected
Line 145: The unit of flowrate “sccm” is not SI unit and I advise to convert to cm3/s or write it as cm3/min.
Author Response: The sccm was changed on cm3/min
Line 152: Is there a special name for the Ag-coated samples? In line 155 is PCL-Ag and PCL-Ag-24h are used). Is the name PCL-COOH necessary?
Author Response: The name PCL-COOH corresponds to sample PCL-ref after plasma treatment, it doesn’t contain silver
Line 158-160: Is Pt coating necessary for PCL-Ag since the surface is conductive with Ag coating? For PCL I would not coat with Pt.
Author Response: Polymer fibers are highly charged when shooting at high magnification or when taking element distribution maps. Magnetron sputter coating technology is one of the approaches to decrease surface charge. The deposition uniformly thin coating less than 2 nm don’t obscure surface details of nanometer dimensions and pretend the accumulated charge Please see : https://doi.org/10.1016/0032-3861(95)90924-Q
- Line 161: “80-mm2” Is this mm2? Is EDX analysis possible for Pt coating?
Author Response: We deposited thin layer (less than 2 nm), thus contribution of Pt less than 0.1 at% since depth of EDX analysis is ⁓1µm
- Line 174: Please remove the space in the sample name to be the same as before “PCL-Ag-24*h”
Author Response: The space was removed
Line 179: “boron”? Not Ag?
Author Response: It’s mistake. “boron” was changed to “silver”
Line 184: “recalculated” could be replace by “normalized”
Author Response: “recalculated” was replace by “normalized”
- Line 190: add “we” before “used”
Author Response: The sentence was corrected
Line 194: “107 CFU/mL” or “107 CFU/mL”
Author Response: We used concentration 107 CFU/mL, we corrected it in the text
Line 211: “but surface PCL-Ag is smoother as compared to PCL - ref” How is this seen? PCL-Ag seems to have particles on the surface which is increasing surface roughness.
Author Response: It is mistake, “smoother” was changed to “rougher”
Line 215: remove “How”
Author Response: “How” was removed
- Table 1 and 2: is there a unit for the numbers or is it [%]?
Author Response: It is at.%, measurement units were added in Tables
Line 216: “from the composition fibers at 0.7 at.%.” should be changed to “from the composition fibers by 0.7 at.%.”
Author Response: “from the composition fibers at 0.7 at.%.” was changed to “from the composition fibers by 0.7 at.%.”
Line 221: “confirming the formation of bonding”
Author Response: “confirming the formation of bonding” was changed “confirming the formation bonds”
Line 224: “The significance differ” to “The significant difference”
Author Response: “The significance differ” was changed “The significant difference”
Line 227: “unbonded NPs were delayed.” To “unbonded NPs were released”
Author Response: “unbonded NPs were delayed” was changed “unbonded NPs were released”
Review English language. Some sentence need to be shortened.
· Line 238: “27,28”
Author Response: The references were corrected
Line 243: unit missing for “533.3”
Author Response: The unit was added “533.3 eV”
Line 261: “After 6 hours of incubation, we observed a 100% reduction in the number of E.coli U20 bacteria (as shown in Fig. 4a)” For E.coli is no value at 6h.
Author Response: The sentence was corrected “After 6 hours of incubation, we observed a 100% antibacterial activity, no value of E.coli U20 bacteria (as shown in Fig. 4a) and Candida auris CBS10913 (as shown in Fig. 5a) were detected”
Figure 4/5: image and caption uses “ref” instead of “PCL-ref” and “Ag” instead of “PCL-Ag”
Author Response: We changed fig4,5 according to recommendation
- Figure 4c/5A has no data for PCL-Ag.
Author Response: The lack of values for the sample PCL-Ag is due to its 100% antibacterial activity, after 6 and 24 hours no bacteria were detected for the sample PCL-Ag
Figures 4/5: Is this for 6hours? The colors of A and C should match.
Author Response: We change color of graphs correspond for results of biofilm formation
Line 267-271: What is the limit of CFU for biofilm formation?
Author Response: Biofilm formation is a complex dynamic process and depends on many physiological, biochemical, and physical factors.
There are no biofilm-specific standards for the efficacy testing of disinfectants (Richter, A.M.; Konrat, K.; Osland, A.M.; Brook, E.; Oastler, C.; Vestby, L.K.; Gosling, R.J.; Nesse, L.L.; Arvand, M. Evaluation of Biofilm Cultivation Models for Efficacy Testing of Disinfectants against Salmonella Typhimurium Biofilms. Microorganisms 2023, 11, 761. https://doi.org/10.3390/microorganisms11030761)
Line 303-309: More qualitative comparison with literature is needed. Better than which state-of-the-art? ref e.g. “Silver-nanoparticle-Incorporated composite nanofibers for potential wound-dressing applications”, Poornima Dubey, Bharat Bhushan, Abhay Sachdev, Ishita Matai, S. Uday Kumar, P. Gopinath, First published: 06 June 2015 https://doi.org/10.1002/app.42473 has similar release characteristics and antibacterial activity.
Author Response: We want to thank the reviewer for suggested great papers. The following paragraphs was added in the Discussion section
Dubey et al. [35] described a new approach for synthesis of Ag NPs with an average size of 15-20 nm based on mixing high-molecular-weight polyethylene oxide (PEO) with AgNO3 solution, followed by the formation of electrospun nanofibers PCL:PEO (1:4 wt.%) with incorporation of 1-3 wt.% Ag NPs, and tested its antibacterial activity. They demonstrated that incorporation of Ag NPs in blended PCL:PEO nanofibers leads to decreasing fiber diameter to 70–150 nm vs. 150–300 nm for pure PCL:PEO fibers, which can be explained by the changing conductivity of the formation solution. The same effect of decreasing diameter of NFs with increasing concentration of Ag was detected by Annur et al. in composite fibers (CS:PEO:Ag NPs). [X] All composite samples (PCL:PEO/Ag NPs) demonstrated high antimicrobial activity, suppressing the growth of antibiotic-resistant GFP E. coli bacteria. They detected the fast-release Ag ions in the first 20 h (40% of loading), followed by a slowly increasing concentration of Ag in the next 76 h.
Annur et al. [36] used Ar plasma to obtain chitosan (CS):PEO:Ag NPs composite nanofibers. On the first step, the nanofibers were produced from the mixture CS:PEO:AgNO3 (5.4:0.6, varying AgNO3 concentration from 0.5 to 2 wt%, respectively). Plasma treatment was carried out using an RF power of 200 W, a chamber pressure of 200 mTorr, and an argon flow of 32 cm3/min. The optimum time for plasma treatment was determined to be 1.5 min. To increase the stability of the samples, after plasma treatment, they were immersed in an aqueous TPP solution. The average size of Ag NPs was determined to be 1.5 nm. The addition of Ag increases the antimicrobial properties of prepared materials seven times against E. coli bacteria.
Foroushani et al. [37] synthesized silk-chitosan nanofibers loaded with curcumin and Ag NPs for wound healing applications. They demonstrated that functional groups of polymers (amino-groups of chitosan and hydroxyl groups of curcumin) bonded Ag ions, forming a polymeric network with a uniform distribution of silver. However, the excellent antibacterial properties of synthesized material against E. coli and S. aureus bacteria are summarized by the activity of Ag NPs, curcumin, and chitosan.
Dubey, P.; Bhushan, B.; Sachdev, A.; Matai, I.; Uday Kumar, S.; Gopinath, P. Silver-Nanoparticle-Incorporated Composite Nanofibers for Potential Wound-Dressing Applications. J. Appl. Polym. Sci. 2015, 132, 1–12, doi:10.1002/app.42473.
Annur, D.; Wang, Z.K.; Liao, J. Der; Kuo, C. Plasma-Synthesized Silver Nanoparticles on Electrospun Chitosan Nanofiber Surfaces for Antibacterial Applications. Biomacromolecules 2015, 16, 3248–3255, doi:10.1021/acs.biomac.5b00920.
Heydari Foroushani, P.; Rahmani, E.; Alemzadeh, I.; Vossoughi, M.; Pourmadadi, M.; Rahdar, A.; Díez-Pascual, A.M. Curcumin Sustained Release with a Hybrid Chitosan-Silk Fibroin Nanofiber Containing Silver Nanoparticles as a Novel Highly Efficient Antibacterial Wound Dressing. Nanomaterials 2022, 12, doi:10.3390/nano12193426.
· Line 321-323: Reference?
Line 330: “simple, eco-friendly, and easily scalable method” Is the method of plasma coating really more simple than one-pot electrospinning and how is it scalable?
Author Response: Indeed, scalable of plasma treatment isn’t simple and easily. We changed the text to “eco-friendly and scalable method”
Line 333:, 417/418 “The polymer layer deposited through” I think plasma treatment activates COOH in a surface layer of PCL but there is no deposition of polymer on the polymeric electrospun PCL fibers.
Author Response: In the present study we used previously developed method of deposition polymerized in plasma COOH-contained thin films on the surface of samples described in 10.1016/j.apsusc.2017.11.174. The plasma deposited coating produced due to the interactions between CO2 and C2H4 gases and surface of PCL fibers in the presence of plasma.

Round 2
Reviewer 2 Report
English has been slightly improved in the revised version, editing of english language is still required
English has been slightly improved in the revised version, editing of english language is still required